# Mechanisms of urate transport and uricosuric drugs inhibition in human URAT1

Wenjun Guo[1,2,3,4,6], Miao Wei[1,6], Yunfeng Li[5], Jiaxuan Xu[1], Jiahe Zang[1], Yuezhou Chen[2,3,5] & Lei Chen[1,2,3,4] ✉

High urate levels in circulation lead to the accumulation of urate crystals in joints and ultimately inflammation and gout. The reabsorption process of urate in the kidney by the urate transporter URAT1 plays a pivotal role in controlling serum urate levels. Pharmacological inhibition of URAT1 by uricosuric drugs is a valid strategy for gout management. Despite the clinical significance of URAT1, its structural mechanism and dynamics remain incompletely understood. Here, we report the structures of human URAT1 (hURAT1) in complex with substrate urate or inhibitors benzbromarone and verinurad at resolution ranges from 3.0 to 3.3 Å. We observe urate in the central substrate-binding site of hURAT1 in the outward-facing conformation and urate is wrapped in the center of hURAT1 by five phenylalanines and coordinated by two positively charged residues on each side. Uricosuric compounds benzbromarone and verinurad occupy the urate-binding site of hURAT1 in the inward-facing conformation. Structural comparison between different conformations of hURAT1 reveals the rocker-switch-like mechanism for urate transport. Benzbromarone and verinurad exert their inhibitory effect by blocking not only the binding of urate but also the structural isomerization of hURAT1.

Urate is the end product of purine catabolism in humans[1]. The majority of urate produced in the human body is excreted into filtrate through the kidneys, and more than 90% of the excreted urate is reabsorbed back into the blood at the proximal convoluted tubules[1]. Therefore, the reabsorption process of urate in the kidney plays an important role in determining the urate level in serum. The abnormal accumulation of urate in serum leads to the precipitation of sodium urate crystals in joints, subsequent inflammation, and ultimately, the development of gout[2]. The reabsorption of urate in the kidney is mediated by urate transporters, among which urate transporter 1 (URAT1) plays an essential role, evidenced by the fact that loss-of-function mutations in hURAT1 lead to idiopathic renal hypouricemia in humans[3]. As such, inhibition of hURAT1 provides a validated therapeutic approach to lower urate levels in serum for gout management. Several hURAT1 inhibitors, including benzbromarone and lesinurad, have been approved for the treatment of gout[4]. In addition, several compounds targeting hURAT1 are in clinical trials against gout[5].

URAT1 (SLC22A12) is a member of the SLC22 major facilitator superfamily transporters. It is primarily expressed in the apical membrane of renal proximal tubule cells[3] and exchanges intracellular anions for extracellular urate[3]. Previous mutagenesis studies have suggested the regions of hURAT1 that bind to urate and uricosuric drugs[6–9]. Recent structures determination of several SLC22 family members have shown the general architecture of this transporter family[10–18]. However, how hURAT1 transports urate and how this process is inhibited by uricosuric drugs remain largely elusive at the

[1]State Key Laboratory of Membrane Biology, College of Future Technology, Institute of Molecular Medicine, Peking. University, Beijing Key Laboratory of Cardiometabolic Molecular Medicine, Beijing 100871, China. [2]Academy for Advanced Interdisciplinary Studies, Peking University, Beijing 100871, China. [3]Peking-Tsinghua Center for Life Sciences, Peking University, Beijing 100871, China. [4]National Biomedical Imaging Center, Peking University, Beijing 100871, China. [5]MOE Key Laboratory of Cell Proliferation and Differentiation, School of Life Sciences, Peking University, Beijing 100871, China. [6]These authors contributed equally: Wenjun Guo, Miao Wei. ✉e-mail: chenlei2016@pku.edu.cn

molecular level. Here, we present structures of hURAT1 in the outward-facing urate-bound conformation and the inward-facing conformation in complex with uricosuric compounds benzbromarone or verinurad. These structures reveal the mechanism of hURAT1 transport and inhibition.

## Results

### Structural determination of hURAT1

Due to the low expression level and poor protein stability of the wild-type hURAT1 (hURAT1$_{WT}$), we sought to improve its biochemical behavior through the consensus design strategy[19]. During the selection of sites for mutations, we aimed to retain the transporter activity for urate and the sensitivity to inhibitors. Therefore, we intentionally avoided sites near the central vestibule of hURAT1 since the center of hURAT1 was predicted to harbor the binding site for urate and inhibitors[6–9]. We also excluded sites on the extracellular protrusion of hURAT1 to preserve the antibody epitopes for detection of surface expression. The resulting construct hURAT1$_{EM}$, contains 15 mutations among the 553 residues of hURAT1, which are distributed in the periphery of hURAT1 (Supplementary Figs. 1 and 2). hURAT1$_{EM}$ exhibits 97% sequence identity to hURAT1$_{WT}$ and demonstrates markedly enhanced urate transport activity compared to hURAT1$_{WT}$ (Fig. 1a). hURAT1$_{EM}$ shows slightly higher Km for urate compared to wild-type (Fig. 1b). Additionally, hURAT1$_{EM}$ displays a higher total expression level, a better peak profile in size-exclusion chromatography, and markedly improved thermostability relative to hURAT1$_{WT}$ (Supplementary Fig. 1b, c). To investigate whether these mutations enhance the surface expression of hURAT1, we developed a monoclonal antibody, Fab15, which targets an extracellular epitope of hURAT1 (Supplementary Fig. 1d). Our findings indicate that hURAT1$_{EM}$ has a higher surface expression level than hURAT1$_{WT}$ (Fig. 1c), partially accounting for its improved transport activity. Furthermore, hURAT1$_{EM}$ is sensitive to inhibition by benzbromarone and verinurad, albeit with slightly higher IC$_{50}$ values compared to hURAT1$_{WT}$ (Fig. 1d, e). This modest decrease in inhibitor potency may be attributed to the elevated surface expression level of hURAT1$_{EM}$ (Fig. 1f), at least in part. Collectively, these results suggest that hURAT1$_{EM}$ effectively recapitulates the functional characteristics of hURAT1$_{WT}$ and can serve as a valuable surrogate for subsequent structural characterization. The purified hURAT1$_{EM}$ proteins in the presence of different ligands were reconstituted into nanodiscs for single-particle cryo-EM studies (Supplementary Fig. 1e, f). The resulting maps reached 3.0–3.3 Å resolutions (Supplementary Figs. 3-7 and Supplementary Table 1), which enabled model building and structural analysis.

### The urate binding site on hURAT1

The cryo-EM structures show that hURAT1 has the common architecture of SLC22 family transporters, which is formed by the N-terminal domain (NTD) with an extracellular protrusion and the C-terminal domain (CTD) (Fig. 1e-h). In the presence of substrate urate, hURAT1 predominantly adopts the outward-facing conformation, in which regions of the NTD and CTD in the inner leaflet of the plasma membrane bind to each other to seal the intracellular gate (Fig. 2). Urate binds in the center of the transmembrane domain of hURAT1 (Fig. 2b, c). The three protrusions on the electron density match the three oxygens on urate, unambiguously defining the binding pose of urate. The xanthine core of urate is wrapped by a hydrophobic cage consisting of five phenylalanines, including F241 of the NTD and F360, F364, F365, and F449 of the CTD (Fig. 2d, e). To probe the functional importance of these five phenylalanines, we mutated them into alanines individually. We found that F241A and F360A mutations enhanced the surface expression of hURAT1 but reduced the urate uptake activity (Fig. 2f, g). F365A, F364A, and F449A mutations not only reduced the surface expression of hURAT1 but also significantly reduced the urate uptake activity, with the near-complete loss-of-

function phenotype for F364A and F449A (Fig. 2f, g). Our results are in agreement with previous findings showing that the F365A mutation increased the Km value of hURAT1[8]. In addition to the binding by these five phenylalanines, the negatively charged 2′-oxygen of urate (pKa -5.4)[20] is coordinated by R477 on M11 and the 8′-OH of urate binds K393 on M8 (Fig. 2d, e). This binding mode of urate is consistent with previous data showing that mutations of either K393 or R477 into alanine, aspartate, or glutamate would significantly increase the Km of urate and decrease the Kcat of hURAT1[8,9]. Moreover, both R477S[21] and R477H[22–24] have been found in idiopathic renal hypouricemia patients, suggesting that R477 of URAT1 is a hotspot for loss-of-functional mutations in humans. Above the urate-binding site, the side chain of M36 partially blocks the extracellular entrance pathway (Fig. 2b), suggesting the current structure might represent the outward-facing partially occluded conformation.

### The binding mode of benzbromarone

Benzbromarone is a high-affinity hURAT1 inhibitor, which was clinically used for the treatment of gout[25]. We determined the structure of hURAT1 in complex with benzbromarone to a resolution of 3.0 Å (Supplementary Figs. 5 and 6). In the presence of benzbromarone, the structure exhibits an inward-facing conformation, in which the extracellular vestibule is sealed and the intracellular vestibule is open. Benzbromarone binds at the urate-binding site (Fig. 3a, b). The ethyl-benzofuran group of benzbromarone is surrounded by C32, S35, M36, F241, and H245 of the NTD, and F360, F364, and F365 of the CTD (Fig. 3c). The dibromo-hydroxyphenyl group of benzbromarone forms hydrophobic interactions with F449 (Fig. 3c). Moreover, the side chains of Q473 and N237 form polar interactions with the hydroxyl group and bromine atom of benzbromarone, respectively (Fig. 3c). These structural observations are in agreement with previous data showing mutations, including S35N, F365Y, and F449Y, would decrease the affinity of benzbromarone[6,7]. To investigate the functional significance of residues interacting with benzbromarone, we mutated them to alanines or their corresponding residues from other OATs (Supplementary Fig. 2), creating the M36T, N237A, F241S, F360T, F364Y, and Q473A mutants. The M36T, F241S, and Q473A mutations significantly reduced urate uptake activity, preventing confident analysis of benzbromarone inhibition (Supplementary Fig. 6d, e). In contrast, the N237A, F360T, and F364Y mutants maintained urate uptake activity similar to hURAT1$_{WT}$ (Fig. 3d, e), but all reduced the potency of benzbromarone to varying degrees. Notably, F360T had the most pronounced effect, highlighting the critical role of F360 in benzbromarone's potency (Fig. 3f).

### The binding mode of verinurad

Verinurad is a high-affinity analog of lesinurad[7]. The map of hURAT1 in the presence of verinurad was resolved to a resolution of 3.2 Å (Supplementary Fig. 7). The structure of hURAT1 in complex with verinurad shows an inward-facing conformation, and is overall similar to that with benzbromarone, with an RMSD of 0.95 Å (Supplementary Fig. 8a). Verinurad also binds to the urate-binding site (Fig. 4a, b). Detailed structural alignment at the drug-binding site reveals a highly similar structure between the benzbromarone-bound state and the verinurad-bound state, except that the rotamer conformation of F241 is distinct (Supplementary Fig. 8b). The naphthalene group of verinurad stacks with F241 on one side and forms hydrophobic interactions with F360, F364, and F449 (Fig. 4b). The cyano group of verinurad forms polar interactions with K393. The pyridine group of verinurad forms a T-shaped π-π interaction with F365, a polar interaction with S35, and hydrophobic interactions with M214. The carboxyl group of verinurad binds Q437 and R477 (Fig. 4b). These structural observations are in agreement with previous data showing that S35N, F241Y, F365Y, and R477K would decrease the potency of verinurad[7], while F449Y and K393R shows little influence on the potency of verinurad[7,9]. To further

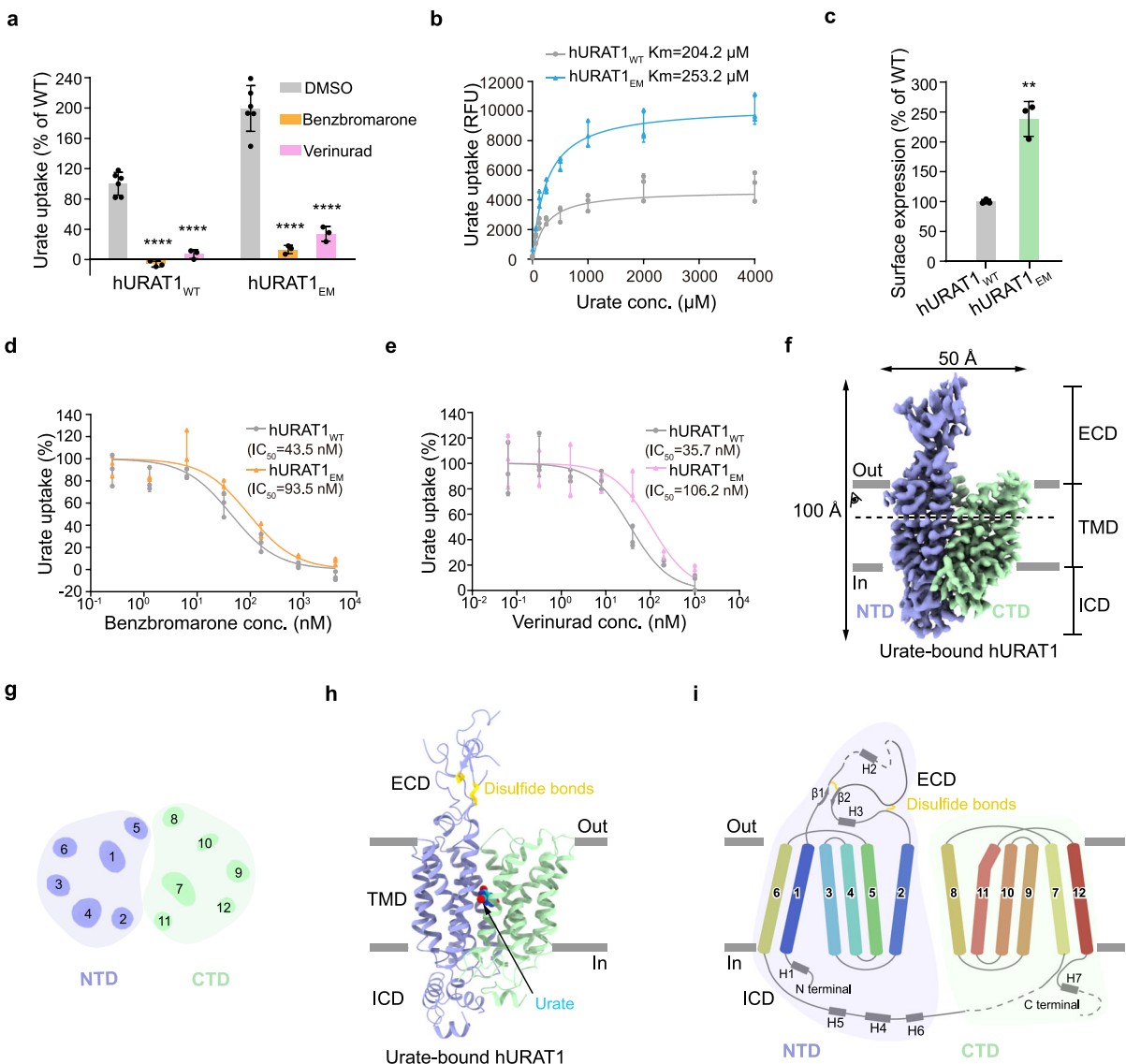

**Fig. 1 | Cryo-EM structure of hURAT1. a** Inhibition of the urate uptake activity of wild-type hURAT1 (hURAT1$_{WT}$) and cryo-EM construct (hURAT1$_{EM}$). In this assay, 4 µM benzbromarone or 1 µM verinurad was used to inhibit urate uptake. Data are normalized to hURAT1$_{WT}$, shown as mean ± s.d.; DMSO, $n = 6$ technical replicates; benzbromarone or verinurad, $n = 3$ technical replicates. The experiment was performed independently twice with similar results. Statistical significance compared with DMSO was determined using ordinary one-way ANOVA with Dunnett's multiple comparisons test. ****$P < 0.0001$. **b** Concentration-dependent uptake of urate by hURAT1$_{WT}$ and hURAT1$_{EM}$. Data are shown as means ± s.d.; $n = 3$ technical replicates. RFU, relative fluorescence units. The experiment was performed independently three times with similar results. **c** Surface expression of hURAT1$_{EM}$. Data are normalized to hURAT1$_{WT}$, shown as means ± s.d.; $n = 3$ technical replicates. The experiment was performed independently three times with similar results. Two-tailed Student's $t$-test. **$P < 0.01$. $P$ value of the normalized surface expression between hURAT1$_{WT}$ and hURAT1$_{EM}$ is 0.0012. **d** Dose-dependent inhibition of hURAT1$_{WT}$ and hURAT1$_{EM}$ by benzbromarone. Data are shown as mean ± s.d.; $n = 3$ technical replicates. The experiment was performed independently twice with similar results. **e** Dose-dependent inhibition of hURAT1$_{WT}$ and hURAT1$_{EM}$ constructs by verinurad. Data are shown as mean ± s.d.; $n = 3$ technical replicates. The experiment was performed independently twice with similar results. **f** Cryo-EM density map of urate-bound hURAT1. Densities are contoured at 8.6 σ. The N-terminal domain (NTD) is shown in purple, and the C-terminal domain (CTD) is shown in green. ECD, extracellular domain; ICD, intracellular domain; TMD, transmembrane domain. **g** Top view of the cross-section of the TMD of hURAT1 at the approximate level indicated by the dashed line and viewing angle denoted in Fig. 1f. The numbers of transmembrane helices are labeled above. **h** Cartoon representation of the urate-bound hURAT1, colored the same as in Fig. 1f. Urate is shown as cyan spheres. Disulfide bonds are shown as golden sticks. **i** Topology of hURAT1. The NTD and CTD are represented as light purple and light green. Unresolved regions are shown as dashed lines. Disulfide bonds are shown as golden sticks.

explore the role of specific residues, we mutated F364 to leucine, and M214, F360, F449, and Q473 to their counterparts in other OATs (Supplementary Fig. 2), creating F364L, M214S, F360T, F449L, and Q473A mutants. The urate uptake activities of F364L, F449L, and Q473A were significantly lower than that of hURAT1$_{WT}$ (Supplementary Fig. 7j, k) and could not be further analyzed. However, F360T and M214S maintained URAT1 uptake activity (Fig. 4c, d). Additionally, the M214S mutation did not affect the potency of verinurad, whereas F360T

substantially reduced it, underscoring the crucial role of the hydrophobic interaction between F360 and verinurad's naphthalene group (Fig. 4e), similar to its important role in benzbromarone inhibition.

## Conformational changes of hURAT1 between states

The structures of hURAT1 in both the outward-facing conformation (urate-bound state) and the inward-facing conformation (benzbromarone/verinurad-bound state) enable the understanding of

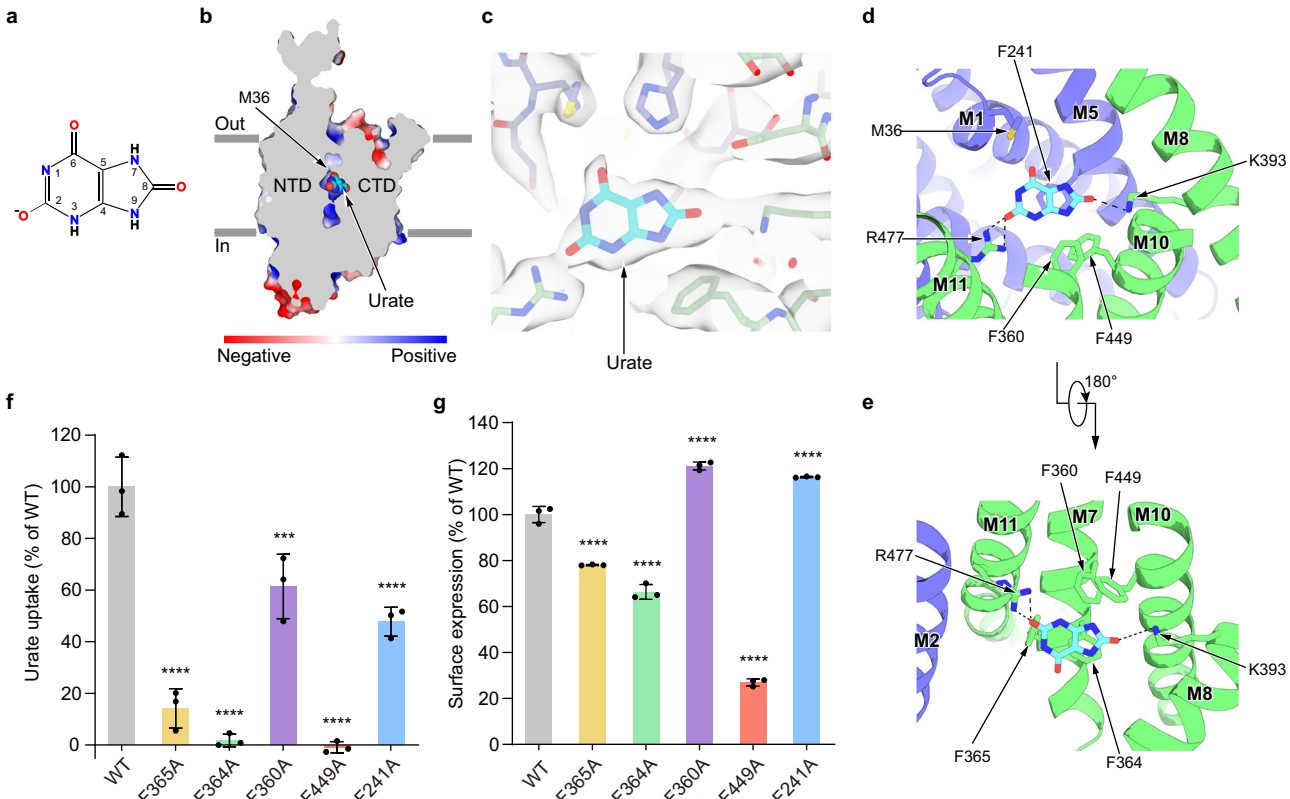

**Fig. 2 | The urate-binding site of hURAT1. a** The chemical structure of urate at a physiological pH. **b** A cut-open view of the urate-binding site of hURAT1. The surface of hURAT1 is colored by the electrostatic potential calculated using Pymol. Urate is shown as cyan spheres. **c** Electron densities of urate and nearby residues. The map is shown as grey surface, contoured at 5 σ. Urate is shown as cyan sticks. The nearby residues are colored the same as in Fig. 1f. **d, e** Interactions between urate and hURAT1. Urate and its interacting residues are shown as sticks. Putative polar interactions are depicted as black dashed lines. **f** Urate uptake activities of various hURAT1 mutants. Data are normalized to hURAT1_{WT}, shown as means ± s.d.; $n = 3$ technical replicates. The experiment was performed independently twice

with similar results. Statistical significance compared with hURAT1_{WT} was determined using ordinary one-way ANOVA with Dunnett's multiple comparisons test. ***$P < 0.001$, ****$P < 0.0001$. $P$ value of normalized uptake efficiency between hURAT1_{WT} and the mutant F360A is 0.0003. **g** Surface expression of various hURAT1 mutants. Data are normalized to hURAT1_{WT}, shown as means ± s.d.; $n = 3$ technical replicates. The experiment was performed independently twice with similar results. Statistical significance compared with hURAT1_{WT} was determined using ordinary one-way ANOVA with Dunnett's multiple comparisons test. ****$P < 0.0001$.

conformational changes of hURAT1 during urate transport (Fig. 5, Supplementary Movie 1). Structural alignment using the larger NTD shows that the CTD moves like a rocker-switch through a rigid-body rearrangement. In the outward-facing conformation, the lower regions of the NTD and CTD bind together to form the cytoplasmic gate below the urate-binding site (Fig. 5b). Particularly, F449 in the CTD blocks urate release through the cytoplasmic exit (Fig. 5b). M222 and M234 of the NTD and I454 of the CTD further seal the cytoplasmic gate (Fig. 5b). On the cytosolic side of hURAT1, we found that E223 on M4 and R284 on H4 of the NTD made electrostatic interactions with R465 on M11 of the CTD and D525 on the loop between M12 and H7, respectively (Fig. 5d, e). These interactions are disrupted in the inward-facing conformation (Fig. 5f). To evaluate the functional importance of the observed polar interactions on the transporting activity of hURAT1, we mutated them into alanines to disrupt the interactions. We found that R465A and R284A mutants dramatically decreased the urate uptake activity of hURAT1, although with slightly decreased surface expression level (Fig. 5h, i), suggesting that both R465-E223 and R284-D525 interaction pairs play critical roles in stabilizing hURAT1 in the outward-facing conformation.

In the inward-facing conformation, the upper regions of the NTD and CTD bind to each other to close the extracellular entrance (Fig. 5). Above the urate-binding site, M36 of the NTD and F364 and L369 of the CTD seal the extracellular gate (Fig. 5c). On the extracellular side of hURAT1, we found that K145 and Q149 on M2 of the NTD form polar

interactions with D370 on M7 and R487 on M11 of the CTD, respectively (Fig. 5f, g). In contrast, these residues do not interact in the outward-facing conformation (Fig. 5d). We found that although R487A and K145A mutants did not decrease the uptake activity of hURAT1, R487A and K145A double mutant markedly reduced urate uptake activity of hURAT1, suggesting that either K145-D370 or Q149-R487 is sufficient to stabilize the inward-facing conformation of hURAT1.

## Inhibition mechanism of hURAT1

In NTD-aligned structures, we found that urate is well accommodated in its binding site in the inward-facing conformation (Supplementary Fig. 8c). Therefore, rigid-body-like motions between the NTD and CTD could sequentially open and close the extracellular gate and cytoplasmic gate, allowing the capture of urate from the kidney filtrate and the subsequent release of urate into the cytosol. In stark contrast, the binding of benzbromarone or verinurad is incompatible with the outward-facing conformation (Supplementary Fig. 8d, e). Specifically, benzbromarone clashes with the side chain of F449 of the CTD, and verinurad clashes with R447 of the CTD of hURAT1 in the outward-facing conformation (Supplementary Fig. 8d, e). These observations are consistent with the fact that urate is the substrate of hURAT1, while benzbromarone and verinurad are inhibitors. They also suggest that benzbromarone and verinurad inhibit hURAT1 not only by competitively occupying the urate-binding site but also by locking hURAT1 in the inward-facing conformation.

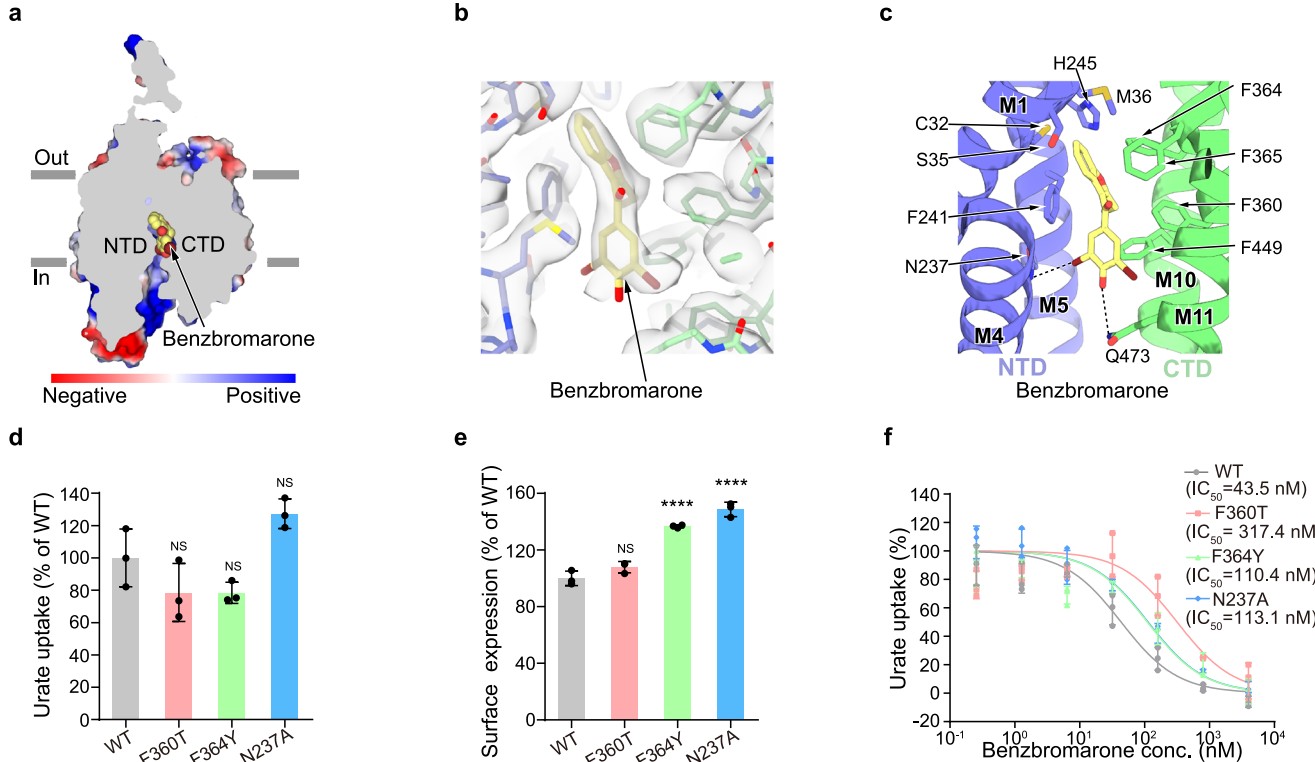

**Fig. 3 | The benzbromarone-binding site of hURAT1. a** A cut-open view of the benzbromarone-binding site of hURAT1. The surface of hURAT1 is colored by the electrostatic potential calculated using Pymol. Benzbromarone is shown as yellow spheres. **b** Electron densities of benzbromarone and nearby residues. The map is shown as grey surface and contoured at 8.2 σ. Benzbromarone is shown as yellow sticks. The nearby residues are colored the same as in Fig. 1f. **c** Interactions between benzbromarone and hURAT1. Benzbromarone and its interacting residues are shown as sticks. Putative polar interactions are depicted as black dashed lines. **d** Urate uptake activities of various mutants of hURAT1. Data are normalized to hURAT1$_{WT}$, shown as means ± s.d.; $n$ = 3 technical replicates. The experiment was performed independently twice with similar results. Statistical significance compared with hURAT1$_{WT}$ was determined using ordinary one-way ANOVA with

Dunnett's multiple comparisons test. ****$P$ < 0.0001; NS, not significant. $P$ value between hURAT1$_{WT}$ and F360T, F364Y, and N237A mutants are 0.2156, 0.2114 and 0.1007, respectively. **e** Surface expression of various mutants of hURAT1. Data are normalized to hURAT1$_{WT}$, shown as means ± s.d.; $n$ = 3 technical replicates. The experiment was performed independently three times with similar results. Statistical significance compared with hURAT1$_{WT}$ was determined using ordinary one-way ANOVA with Dunnett's multiple comparisons test. ****$P$ < 0.0001; NS, not significant. $P$ value of normalized surface expression between hURAT1$_{WT}$ and the mutant F360T is 0.1071. **f** Dose-dependent inhibition of hURAT1$_{WT}$ and various mutants by benzbromarone. Data are shown as means ± s.d.; $n$ = 3 technical replicates. The experiment was performed independently twice with similar results. Data for hURAT1$_{WT}$ is the same as shown in Fig. 1d.

## Discussion

The structures of hURAT1 in both the outward-facing and inward-facing conformations reveal classical rocker-switch-like global conformational changes, which are common to many MFS transporters[26]. Here, we compare the structures of hURAT1 with human OCT1 (hOCT1), because the structures of hOCT1 in multiple conformations are available[12,14,15]. The urate-bound hURAT1 is akin to hOCT1 in complex with spironolactone in the outward-facing partially occluded conformation (PDB ID: 8JTZ), hOCT1 in complex with metformin in outward-open conformation (PDB ID: 8JTS), and hOCT1 in complex with metformin in outward-occluded conformation (PDB ID: 8JTT)[15] (Supplementary Fig. 9a–c). The benzbromarone or verinurad-bound state of hURAT1 is similar to hOCT1 in multiple states, including the apo state (PDB ID:8SC1), bound with diltiazem (PDB ID:8SC2) and bound with fenoterol (PDB ID:8SC3) in inward-open conformation, also bound with metformin (PDB ID:8JTV), spironolactone (PDB ID:8JU0) in inward-occluded conformation[14,15] (Supplementary Fig. 9d–h). The overall conformational similarity between hURAT1 and OCT1 suggests that although SLC22 family members have distinct substrate specificities, they share a common structural mechanism for substrate transport.

The structural rearrangements of hURAT1 are accompanied by the concerted formation and disruption of interdomain interactions and the sequential exposure of the urate-binding site at the center of

hURAT1 to the extracellular side and cytosol. Disrupting these interactions by mutations would decrease the urate uptake activity of hURAT1, likely because they block structural isomerization of hURAT1 by destabilizing either the outward-facing conformation (R284A or R465A mutation) or the inward-facing conformation (K145A and R487A mutation) (Fig. 5). Structural analysis also reveals that bulky inhibitors, including benzbromarone and verinurad, lock hURAT1 in the inward-facing conformation to inhibit the transporting activity of hURAT1.

While this work was under submission, two additional studies were published with associated PDB files available[27,28], enabling a comprehensive comparison. In one study, the structure of the hURAT1 R477S loss-of-function mutant was determined (PDB ID: 8WJG)[28]. This structure is similar to the outward-facing conformation resolved in our study, though the outer region of M7 (residues 362-372) in R477S exhibits outward movement (Supplementary Fig. 10a, b). This suggests that the R477S mutation may preferentially stabilize hURAT1 in the outward-facing conformation. The other study involved the fusion of MBP and DARPin to hURAT1 and replacing the extracellular loop (residues 60–65) and intracellular region (residues 280-343) with that of OAT4, resulting in a chimeric URAT1 (URAT1$_{chi}$) fusion protein, which was used for the determination of multiple structures[27]. The structures of URAT1$_{chi}$ in complex with verinurad in the inward-facing conformation (PDB ID: 9B1I) and in complex with urate in the outward-

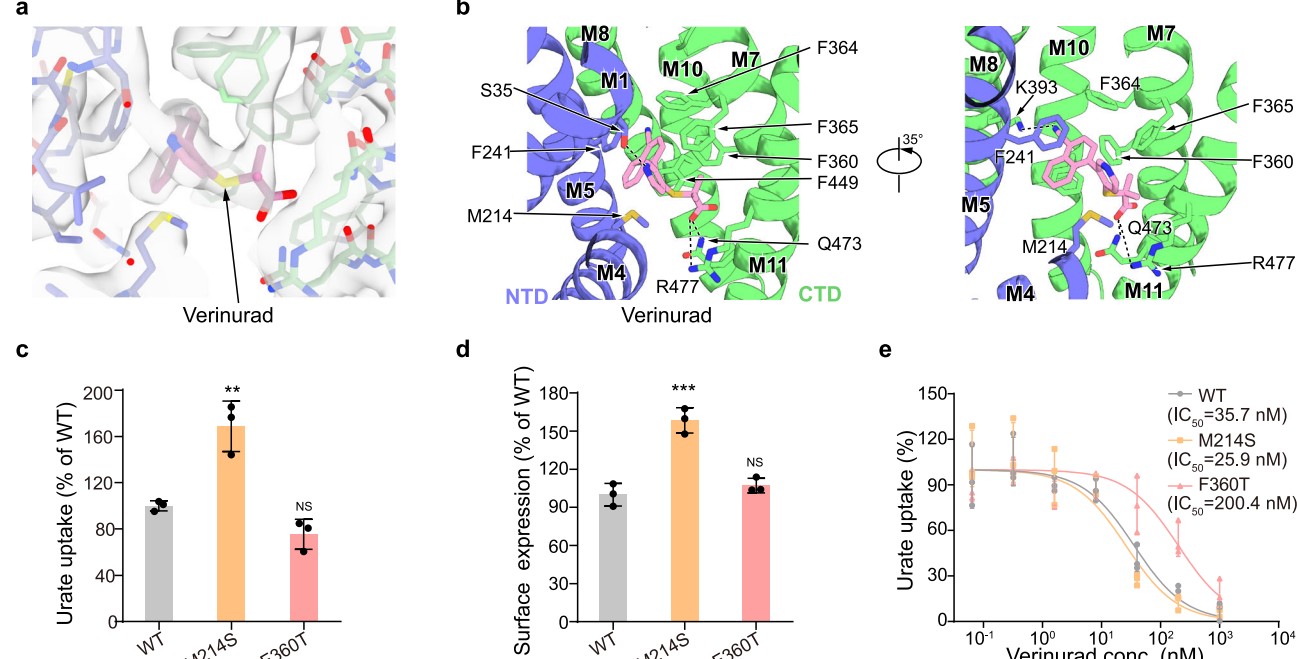

**Fig. 4 | The verinurad-binding site of hURAT1. a** Electron densities of verinurad and nearby residues. The map is shown as grey surface and contoured at 10 $\sigma$. Verinurad is shown as pink sticks. The nearby residues are colored the same as in Fig. 1f. **b** Interactions between verinurad and hURAT1. Verinurad and its interacting residues are shown as sticks. Putative polar interactions are depicted as black dashed lines. **c** Urate uptake activities of various mutants of hURAT1. Data are normalized to $hURAT1_{WT}$, shown as means ± s.d.; $n$ = 3 technical replicates. The experiment was performed independently twice with similar results. Statistical significance compared with $hURAT1_{WT}$ was determined using ordinary one-way ANOVA with Dunnett's multiple comparisons test, **$P$ < 0.01; NS, not significant. $P$ value between $hURAT1_{WT}$ and mutants M214S and F360T are 0.0023 and 0.149,

respectively. **d** Surface expression of various mutants of hURAT1. Data are normalized to $hURAT1_{WT}$, shown as means ± s.d.; $n$ = 3 technical replicates. The experiment was performed independently three times with similar results. Statistical significance compared with $hURAT1_{WT}$ was determined using ordinary one-way ANOVA with Dunnett's multiple comparisons test. ***$P$ < 0.001; NS, not significant. $P$ value of normalized surface expression between $hURAT1_{WT}$ and M214S and F360T mutants are 0.0003 and 0.5123, respectively. **e** Dose-dependent inhibition of $hURAT1_{WT}$ and various mutants by verinurad. Data are shown as means ± s.d.; $n$ = 3 technical replicates. The experiment was performed independently twice with similar results. Data for $hURAT1_{WT}$ is the same as shown in Fig. 1e.

facing conformation (PDB ID: 9B1L) are overall quite similar to our findings. However, there are notable differences in the ligand binding poses (Supplementary Fig. 10c–g). Specifically, there is a rotational difference in the carboxyl group of verinurad, indicating structural dynamics in this region (Supplementary Fig. 10d). More strikingly, the binding poses of urate differ significantly between the two structures, despite the urate-interacting residues being nearly identical (Supplementary Fig. 10f, g). We found that there is a substantial difference in pH between the two studies: they utilized pH 8.5[27], while we used pH 7.4 for our cryo-EM sample preparation, which is much closer to the physiological condition. This variation in pH may influence the protonation state of urate or surrounding residues, potentially leading to the observed differences in urate binding poses.

Collectively, these studies provide a structural framework for understanding the physiology of urate reuptake by hURAT1 and set the foundation for future structure-based drug discovery for gout treatment.

# Methods
## Ethical statement
This research complies with all relevant ethical regulations. Animal studies were approved by the Institutional Animal Care and Use Committee (IACUC) of Peking University and conformed to the Guide for the Care and Use of Laboratory Animals (IMM-ChenL-3).

## Constructs and Cell culture
The cDNA of full-length $hURAT1_{WT}$, $hURAT1_{EM}$, and related mutants were cloned into the N-terminal GFP-tagged pBMCL1 vector[29] for

uptake assay and protein purification. Sf9 insect cells (Thermo Fisher Scientific # 12659017) were cultured in SIM SF (Sino Biological) at 27 °C. FreeStyle 293 F suspension cells (Thermo Fisher Scientific # R79007) were cultured in FreeStyle 293 medium (Thermo Fisher Scientific) supplemented with 1% fetal bovine serum (FBS, VisTech), 67 µg/ml penicillin (Macklin), and 139 µg/ml streptomycin (Macklin) at 37 °C with 6% $CO_2$ and 70% humidity. The cell lines were routinely checked to be negative for mycoplasma contamination but had not been authenticated.

## Consensus design
Consensus design was carried out as previously reported[19]. First, 120 sequences of SLC22A12 family members were downloaded from UniProt and aligned using MUSCLE[30], visualized using JalView[31], and mapped to the predicted structure downloaded from the AlphaFold2 database[32]. We intentionally avoided mutations near the central vestibule of hURAT1 and mutations on the extracellular protrusion of hURAT1. The resulting construct ($hURAT1_{EM}$) contains 15 designed mutations, including A162V, M186S, A195T, A226S, V264A, W294R, W300R, G309R, M330V, P333A, M343T, F348L, M383V, F384L, H402R.

## Mice
BALB/c female mice, aged between 6 and 8 weeks, were used for immunization. Mice were housed in groups of 3 at specific pathogen free (SPF) status with standard 12 h light/12-h dark cycles. Ambient animal room temperature is 23 °C, controlled within ±3 °C and room humidity is 50%, controlled within ±20%. FreeStyle 293 F cells infected by baculovirus carrying $hURAT1_{WT}$ with N-terminal GFP tag for 24 h

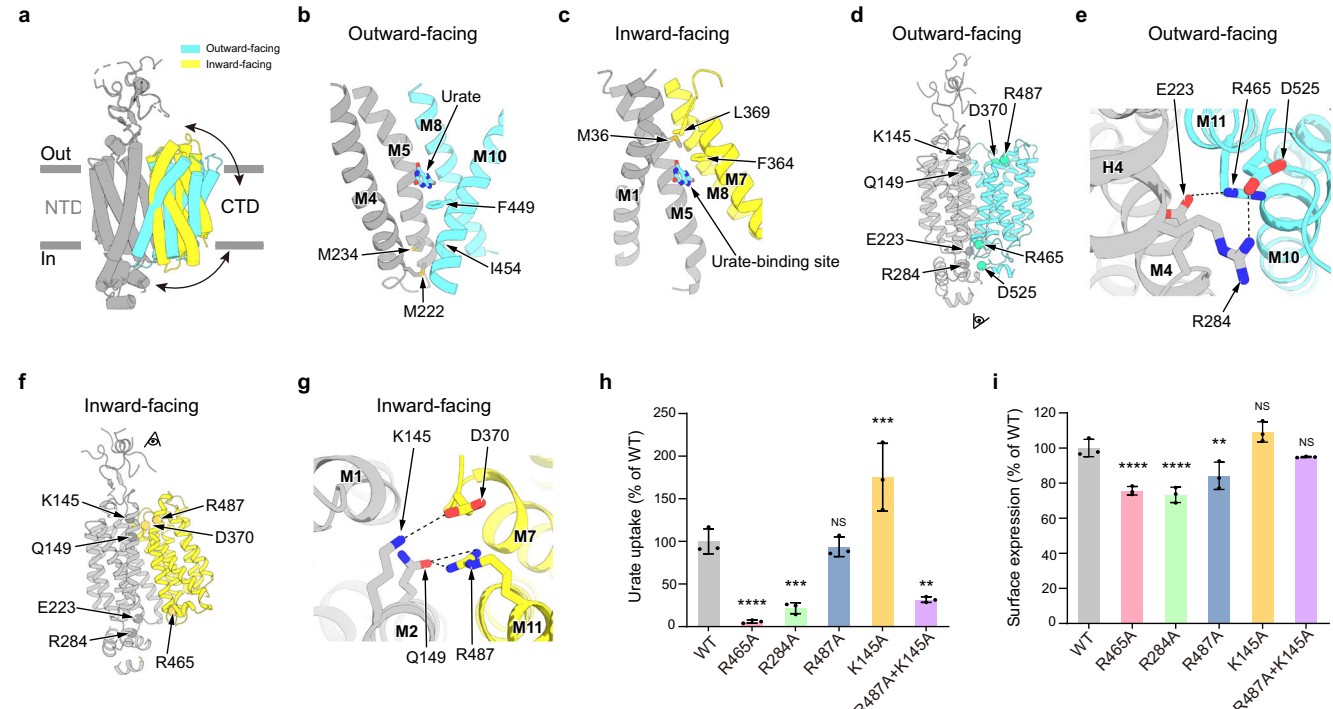

**Fig. 5 | Structural changes of hURAT1 between outward-facing and inward-facing conformations. a** Superposition of the outward-facing (urate-bound, CTD in cyan) and inward-facing (benzbromarone-bound, CTD in yellow) states of hURAT1. The two structures are aligned based on their NTDs (grey). **b** Cytoplasmic gate in the outward-facing state. Key residues are shown as sticks. **c** Extracellular gate in the inward-facing state. Key residues are shown as sticks. **d** Location of key residues (Cα shown as spheres) in the outward-facing state. **e** Bottom view of key polar interactions in the outward-facing state. Putative polar interactions are depicted as black dashed lines. Viewing angle is indicated in Fig. 5d. **f** Location of key residues (Cα shown as sphere) in the inward-facing state. **g** Top view of key polar interactions in the inward-facing state. Putative polar interactions are depicted as black dashed lines. Viewing angle is indicated in Fig. 5f. **h** Urate uptake activities of various mutants involved in polar interactions. Data are normalized to hURAT1$_{WT}$, shown as means ± s.d.; $n$ = 3 technical replicates. The experiment was

performed independently twice with similar results. Statistical significance compared with hURAT1$_{WT}$ was determined using ordinary one-way ANOVA with Dunnett's multiple comparisons test. **$P < 0.01$, ***$P < 0.001$, ****$P < 0.0001$; NS, not significant. $P$ value of normalized uptake efficiency between hURAT1$_{WT}$ and mutants R284A, R487A, K145A and R487A + K145A are 0.0003, 0.9935, 0.004 and 0.001, respectively. **i** Surface expression of various mutants involved in conformational changes. Data are normalized to hURAT1$_{WT}$, shown as means ± s.d.; $n$ = 3 technical replicates. The experiment was performed independently twice with similar results. Statistical significance compared with hURAT1$_{WT}$ was determined using ordinary one-way ANOVA with Dunnett's multiple comparisons test. **$P < 0.01$, ****$P < 0.0001$; NS, not significant. $P$ value of normalized surface expression between hURAT1$_{WT}$ and R487A, K145A and R487A + K145A mutants are 0.0039, 0.1113 and 0.5737, respectively.

were collected at 150 $g$ for 5 min and washed with PBS twice. Synthetic oligodeoxynucleotides containing unmethylated CpG motifs (5′-tccatgacgttcctgacgtt-3′) by Azenta Life Science was used as an adjuvant for immunization with hURAT1-expressing cells. The intraperitoneal injection was carried out at two 3-week intervals and followed by a 5-week interval. 10 days after the third immunization, spleens were isolated for B cell sorting. All procedures involving animals followed the protocols approved by the IACUC of Peking University and conformed to the Guide for the Care and Use of Laboratory Animals.

### Biotinylation of hURAT1 for antibody selection

hURAT1$_{EM}$ with an extra N terminal Avi tag and 30×GS linker between GFP and the hURAT1$_{EM}$ sequence was purified in a similar manner as described as follows. The target protein was assembled into the peptidisc on the Streptactin Beads by loading 1 ml 1 mg/ml NSPr in 20 mM Tris pH 8.0[33] and incubating for 1 h. Then the column was washed with 50 ml TBS buffer (20 mM Tris pH 8.0 at 4 °C, and 150 mM NaCl) to remove free NSPr. The assembled peptidiscs were eluted with 5 mM D-desthiobiotin (IBA) in TBS. The eluted protein was concentrated to 40 μM and added home-purified GST-BirA (1 μg GST-BirA per nmol hURAT1$_{EM}$), 10 mM adenosine triphosphate (ATP), 10 mM MgCl$_2$, and biotin (molar ratio of biotin to hURAT1$_{EM}$ is 4:1, Sigma) for 16 h at the cold room. Prescission protease was also added to the system to remove the N-terminal tags of hURAT1$_{EM}$. Biotinylated hURAT1$_{EM}$ was

further purified by Superose 6 increase 10/300 GL column (GE Healthcare) running in a buffer containing 20 mM HEPES (pH=7.4), and 150 mM NaCl. The peak fractions were pooled, supplemented with 40% glycerol, and stored at −20 °C.

### B cell sorting, and single B cell RT-PCR

After depletion of red blood cells (RBCs) by RBC lysis buffer (Thermo Fisher Scientific), splenocytes from immunized mice were washed and incubated with biotinylated 5 μg/ml hURAT1$_{EM}$ in peptidisc on ice for 15 min. Cells were washed to stain PE and APC conjugated streptavidin for double gating hURAT1$_{EM}$-binding cells, and incubated with the following mAbs (from Biolegend) for 10 min at 4 °C in PBS supplemented with 2% FBS: IgG1 (406632, clone: RMG1-1, lot: B422285), CD138 (142519, clone: 281-2, lot: B378402), CD19 (115543, clone: 6D5, lot: B386950), Streptavidin (405207, lot: B380870); and following mAbs from Life Technologies: GL7 (53-5902-82, clone: GL-7 (GL7), lot: 2442288), CD38 (25-0381-82, clone: 90, lot: 2460145), B220 (47-0452-82, clone: RA3-6B2, lot: 2703742), CD4 (69-0041-82, clone: GK1.5, lot: 2629005), CD8a (69-0081-82, clone: 53-6.7, lot: 2446938), IgD (63-5993-82, clone: 11-26c (11-26), lot: 2634835), Streptavidin (12-4317-87, lot: 2514129). Fixable Viability Dye (Biolegend, 423114) was added to separate dead cells. All the antibodies are used at a 1:200 dilution for staining. Single live PE and APC double positive germinal center B cells (CD4⁻CD8a⁻CD19⁺CD138⁻B220⁺CD38⁻GL7⁺) were sorted into 96-well

plates containing 4 µl cell lysis buffer (0.5×PBS, 10 mM DTT, 4 U RNa-seout). The single B cell RT-PCR was performed as described[34]. Briefly, RNA was reverse-transcribed with 50 U maxima reverse transcriptase (Thermo Scientific), 200 ng random primer (hexadeoxyribonucleotide mixture pd(N)6, Takara), 4 U Rnaseout (Thermo Fisher Scientific), 1% Igepal CA-630 (Sigma), and 10 nM dNTP. Subsequently, 2 µl cDNA for each cell was used to amplify heavy, kappa and lambda chain through two rounds of nested PCR[34]. PCR products were sequenced and aligned to germline immunoglobulin V, (D) and J gene segments by IgBlast[34]. Fab fragments were expressed using a modified BacMam expression vector[35] in FreeStyle 293 F suspension cells and purified using 6×His tag at the C-terminus of heavy chain.

## Urate uptake assay
For uptake assays, FreeStyle 293 F cells were cultured in 24-well plates pre-coated with 100 µg/ml poly-lysine. After transfection for 40 h, cells were washed with 800 µl uptake buffer (125 mM Na-gluconate, 4.8 mM K-gluconate, 1.3 mM MgSO$_4$, 1.2 mM Ca-gluconate, 10 mM glucose, and 25 mM HEPES, pH 7.4) per well to remove the culture medium. To measure the potency of inhibitors, 250 µl uptake buffer with different concentrations of inhibitors was preincubated with cells for 10 min. Then, 250 µl 800 µM urate with inhibitors in uptake buffer was added to cells for another 20 min at 37 °C with 5% CO$_2$. Subsequently, cells were washed twice with 800 µl ice-cold uptake buffer per well to stop the uptake reaction and then lysed with 90 µl 1% DDM in 50 mM Tris (pH 7.5 at RT) for 30 min at room temperature. The cell lysates were then centrifuged for 5 min at 17,000 g to remove the debris. The urate accumulated in the cells was measured using an enzymatic assay. In the assay, uricase catalyzes the conversion of urate to allantoin, hydrogen peroxide, and carbon dioxide. In the presence of horseradish perox-idase (HRP), the hydrogen peroxide then reacts stoichiometrically with the Amplex Red reagent to generate the red-fluorescent oxidation product, resorufin. The resorufin fluorescence (excitation, 530 nm; emission, 590 nm) was measured using a microplate reader (Tecan Infinite M Plex). The reactions were performed at 37 °C for 30 min in 100 µl of 50 mM Tris (pH 7.5, at RT) buffer, containing an aliquot (50 µl) of the supernatant, 0.04 mg/ml uricase, 0.001 mg/ml HRP, and 0.05 mM Amplex Red. The resorufin fluorescence signals in each well were normalized to the protein concentration determined using the BCA method, which represents cell numbers. Resorufin fluorescence signals of cells transfected with an empty vector and incubated with DMSO were measured as the background signal. The percentage of specific uptake of each well was calculated by subtracting the back-ground from the total signal, and then normalized to the specific uptake only with DMSO for the same construct. Nonlinear regressions were fitted using log(inhibitor)vs. response equation: $Y = 100/(1 + 10^{((X-LogIC50)))}$ to determine the IC$_{50}$ of inhibitors with GraphPad Prism v.6.

To study the uptake efficiency of residues related to urate binding and residues related to conformational changes, after transfection for 40 h, the FreeStyle 293 F cells were washed and incubated with 400 µM urate for 10 min, then the cells were processed the same as before mentioned.

To measure the Km of hURAT1$_{WT}$ and hURAT1$_{EM}$, after transfec-tion for 40 h, the FreeStyle 293 F cells were washed and incubated with urate at different concentrations for 10 min. Then the cells were pro-cessed the same as mentioned above. Nonlinear regressions were fit-ted using Michaelis-Menten equation to determine the Km of urate with GraphPad Prism v.6.

## Surface labeling
The surface labeling was carried out similarly as described previously[36]. FreeStyle 293 F cells were plated onto a poly-D-lysine-coated 24-well plate and transfected with hURAT1$_{WT}$, hURAT1$_{EM}$, or related mutants for 40 h. The cells were washed with 500 µl PBS per well twice, fixed using 300 µl 4% formaldehyde in PBS for 30 min, and washed with 500 µl PBS per well twice again. Then, the cells were blocked with 300 µl 3% goat serum in PBS per well for 30 min and labeled with 150 µl 50 nM primary antibody Fab15, selected by B cell sorting, per well for 1 h. After washing with PBS three times, cells were incubated with rabbit anti-HA antibody (3724; CST, diluted 1:2500 in blocking buffer) for another 1 h. After another 3 washes with PBS, cells were incubated with horseradish-peroxidase (HRP) labeled rabbit secondary antibody (31460; Invitrogen, diluted 1:2500 in blocking buffer) for 30 min. After extensive washing, the cells were incubated with High-Sig ECL Western Blotting Substrate (Tanon) for 2 min, and chemiluminescence signals were measured with an Infinite M Plex plate reader (Tecan).

## Thermostability assay
For the thermostability assay, baculovirus-infected cells were solubi-lized with 1% LMNG and 0.1% CHS in TBS buffer for 30 min on ice. Cell lysates from infected cells were centrifuged at 86,600 g for 10 min and then divided into three aliquots. Supernatants of hURAT1$_{WT}$ and hURAT1$_{EM}$ were heated at 48 °C for 10 min. All samples were further centrifuged at 86,600 g for 30 min before the FSEC[37] analysis. The experiments were independently repeated twice for each construct.

## Protein expression and purification
The baculoviruses of N-terminal GFP-tagged hURAT1$_{EM}$ were produced using the Bac-to-Bac system and amplified in Sf9 insect cells. For protein expression, FreeStyle 293 F cells cultured in Freestyle medium at a density of 2.5–3.0 × 10$^6$ cells per ml were infected with 2% volume of P2 virus. After infection for 8–12 h, 10 mM sodium butyrate was added to the culture and cultured at 37 °C for another 36 h before collecting. Cells were collected by centrifugation at 4000 g (JLA 8.1000, Beckman Coulter) for 10 min, and washed with TBS plus 1 µg/ml aprotinin, 1 µg/ml pepstatin and 1 µg/ml leupeptin. The collected cells were broken by sonication in TBS plus 1 µg/ml aprotinin, 1 µg/ml pepstatin and 1 µg/ml leupeptin and 1 mM phenylmethanesulfonyl fluoride (PMSF). Unbroken cells and cell debris were removed by centrifugation at 7500 g for 10 min. The supernatant was centrifuged at 186,400 g for 1 h in Ti45 rotor (Beckman). Membrane pellets were harvested, frozen and stored at −80 °C.

To prepare the benzbromarone-bound hURAT1$_{EM}$, the mem-brane pellets corresponding to a 1-L culture were solubilized in 1% LMNG, 0.1% CHS, 1 µM benzbromarone with protease inhibitors in TBS buffer and stirred at 4 °C for 1 h. The insolubilized materials were removed by centrifugation at 164,700 g for 30 min in Ti70 rotor (Beckman Coulter). The solubilized proteins were loaded onto a 2-ml Streptactin beads 4FF (Smart-Lifesciences) column and washed with 40 ml wash buffer (0.01% LMNG, 0.001% CHS and 1 µM benzbromarone in TBS) plus 10 mM MgCl$_2$ and 1 mM ATP to remove contamination of heat shock proteins. The column was further washed with 20 ml wash buffer (0.08% digitonin and 1 µM benzbromarone in TBS). The target protein was eluted using 50 mM Tris pH 8.0 at 4 °C, 150 mM NaCl, 0.08% digitonin, 1 µM benzbromarone and 5 mM D-desthiobiotin (IBA). The elution from Streptactin beads was diluted six times and loaded onto the HiTrap Q HP column (Smart-Lifesciences) equilibrated with 20 mM Tris pH 8.0 at 4 °C, 0.08% digitonin and 1 µM benz-bromarone. The hURAT1$_{EM}$ was eluted by a linear gradient from 0 mM NaCl to 500 mM NaCl. Fractions of the HiTrap Q HP column containing the hURAT1$_{EM}$ were used for nanodisc reconstitution. The linear NW9 MSP protein was purified as previously described in ref. 38. The hURAT1$_{EM}$ was mixed with NW9 and brain polar lipids extract (BPLE, solubilized in 1% digitonin, Avanti) at a molar ratio of protein:NW9:BPLE = 1:4:200. After incubating the mixture at 4 °C for 1 h, Bio-beads SM2 (Bio-Rad) were added to remove detergent and the mixture was constantly rotated at 4 °C for 1 h to initiate the reconstitution. A second batch of Bio-beads SM2 were

then added, and the sample was incubated at 4 °C overnight with constant rotation. A third batch of Bio-beads SM2 were then added for another 1 h the next day. Bio-beads SM2 were then removed and the reconstitution mixture was cleared by centrifugation at 86,600 g for 30 min in the TLA 100.3 rotor (Beckman Coulter).

The hURAT1$_{EM}$ reconstituted in lipid nanodisc was further purified by Streptactin beads in buffer without detergent to remove excess NW9 protein. To cleave the tags off the hURAT1$_{EM}$ protein, the concentrated elution was incubated with H3CV protease at 4 °C overnight. The mixture was further purified by glutathione Beads (Smart-Lifesciences) to remove the H3CV. Finally, the flow-through from the glutathione Beads column was concentrated and loaded onto a Superose 6 increase 10/300 GL column in TBS buffer that contained 1 μM benzbromarone. The peak fractions corresponding to the monomeric hURAT1$_{EM}$ reconstituted in lipid nanodisc were collected for cryo-EM sample preparation.

For the sample of hURAT1$_{EM}$ in complex with verinurad, the expression and purification process were the same as described above, except 0.1 μM verinurad was added during the purification process.

For the sample of hURAT1$_{EM}$ in complex with urate, the expression and purification process were similar as described above, except 5 mM urate was added during the purification process, and NaCl and Tris-HCl were replaced by 150 mM sodium gluconate and 20 mM HEPES, pH 7.4, respectively.

### Cryo-EM sample preparation
For the urate-bound state, the purified hURAT1$_{EM}$ in nanodisc was concentrated to 15.8 mg/ml. For the benzbromarone-bound state, the purified hURAT1$_{EM}$ in nanodisc was concentrated to A$_{280}$ = 20 and supplemented with 500 μM benzbromarone. For the verinurad-bound state, the purified hURAT1$_{EM}$ in nanodisc was concentrated to A$_{280}$ = 10 with an estimated concentration of 7.9 mg/ml and supplemented with 500 μM verinurad. To avoid the preferred orientation, 3 mM fluorinated fos-choline-8 (Anatrace) was added to the sample before cryo-EM sample preparation. Holey carbon grids (Quantifoil Au 300 mesh, R 0.6/1) were glow-discharged by Solarus advanced plasma system (Gatan). Aliquots of 2.5 μl concentrated protein sample were applied on glow-discharged grids and the grids were blotted for 3.5 s before being plunged into liquid ethane using Vitrobot Mark IV (Thermo Fisher Scientific).

### Cryo-EM data collection
The prepared cryo-grids were first screened on a Talos Arctica electron microscope (Thermo Fisher Scientific) operating at 200 kV with a K2 camera (Gatan). The screened grids were subsequently transferred to a Titan Krios electron microscope (Thermo Fisher Scientific) operating at 300 kV with a K3 camera (Gatan) and a GIF Quantum energy filter (Gatan) set to a slit width of 20 eV. Images were automatically collected using EPU-2.12.1.2782REL (Thermo Fisher Scientific) in super-resolution mode at a nominal magnification of 81,000×, corresponding to a calibrated super-resolution pixel size of 0.5335 Å with a preset defocus range from −1.5 to −1.8 μm. Each image was acquired as a 3.73 s movie stack of 47 frames with a dose rate of 21.46 e$^-$/pixel/s, resulting in a total dose of about 70 e$^-$/Å$^2$.

### Cryo-EM image processing
The image processing workflows are illustrated in Supplementary Figs. 3c, 5c and 7d. Super-resolution movie stacks were collected. Motion-correction, two-fold binning to a pixel size of 1.067 Å, and dose-weighting were performed using MotionCor2-1.3.2[39]. The contrast transfer function (CTF) parameters of dose-weighted micrographs were estimated by cryoSPARC-4.5.3[40]. Micrographs with ice or ethane contamination and empty carbon were removed manually. Micrographs with a defocus among −0.7 to

−3 μm and with a resolution better than 4 Å were selected for further data processing. Reference-free 2D classification was performed to remove contaminants. The resulting particles were subjected to 3D classification using the initial models generated using cryoSPARC. The classes that showed obvious secondary structure features were selected and subjected to an additional several rounds of 3D classification. The resulting particles were subjected to non-uniform refinement[41]. To further improve the resolution, seed-facilitated 3D classification[42] was performed with particles picked using Topaz[43]. Several rounds of seed-facilitated 3D classification using good and biased references or references with resolution gradients were performed. The resulting particles were subjected to non-uniform refinement and local refinement in cryoSPARC with the mask of hURAT1, yielding maps with a resolution of 3.34 Å, 2.99 Å, and 3.21 Å for the urate-bound state, benzbromarone-bound state and verinurad-bound state, respectively. All of the resolution estimations were based on a Fourier shell correlation of 0.143 cut-off after correction of the masking effect. Local-resolution maps were calculated using cryoSPARC.

### Model building
The model of hURAT1 was built by the SWISS-MODEL[44] server using rOAT1 (PDB ID: 8SDU)[13] as the template. The initial model was then fitted into the cryo-EM map using UCSF Chimera-1.15[45] and rebuilt manually using Coot-0.9.8.93[46]. Model refinement was performed using phenix.real_space_refine in PHENIX1.19.2[47]. The coordinates and restraints for urate, benzbromarone and verinurad were derived using the eLBOW tool in PHENIX. Images were produced using Pymol-2.6.0, UCSF Chimera-1.15, and ChimeraX-1.2.5[48]. Sequence alignment was performed with PROMALS3D[49] and illustrated using BioEdit-7.2.5.

### Quantification and statistical analysis
Global resolution estimations of cryo-EM density maps are based on the Fourier shell correlation 0.143 criterion[50]. The local resolution was estimated using cryoSPARC. The number of independent experiments (n) and the relevant statistical parameters for each experiment (such as mean and s.d.) are described in the figure legends. No statistical methods were used to predetermine sample sizes.

### Reporting summary
Further information on research design is available in the Nature Portfolio Reporting Summary linked to this article.

## Data availability
The data that support this study are available from the corresponding authors upon request. Cryo-EM maps of the hURAT1 in the urate-bound state, the benzbromarone-bound state and verinurad-bound state have been deposited in the Electron Microscopy Data Bank (EMDB) under the ID codes: EMD-60823 (map of urate-bound state hURAT1); EMD-60824 (map of benzbromarone-bound state hURAT1); EMD-60825 (map of verinurad-bound state hURAT1), respectively. Atomic models of hURAT1 in the urate-bound state, the benzbromarone-bound state and verinurad-bound state have been deposited in the Protein Data Bank (PDB) under the ID codes: 9IRW (atomic model of urate-bound state hURAT1); 9IRX (atomic model of benzbromarone-bound state hURAT1), 9IRY (atomic model of verinurad-bound state hURAT1), respectively. The entry 8SDU used in this study were downloaded from the PDB. Source data are provided with this paper.

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

## Acknowledgements

We thank all of the Chen Lab members for their kindly help. Cryo-EM data collection was supported by the Electron microscopy laboratory and the Cryo-EM platform of Peking University, with the assistance of Xuemei Li, Zhenxi Guo, Changdong Qin, Xiaojuan Hui, and Guopeng Wang. Part of the structural computation was also performed on the Computing Platform of the Center for Life Science and High-performance Computing Platform of Peking University. We thank the National Center for Protein Sciences at Peking University in Beijing, China for assistance with negative stain EM. The work is supported by grants from the Ministry of Science and Technology of China (National Key R&D Program of China, 2022YFA0806504 to L.C.), National Natural Science Foundation of China (32225027 to L.C., 32370942 and 82341049 to Y.C., 32301009 to W.G.), the Center For Life Sciences (CLS to L.C. and Y.C.), Benyuan Young Investigator Award to Y.C. W.G. is supported by the postdoctoral foundation of the Peking-Tsinghua Center for Life Sciences, Peking University (CLS).

## Author contributions

L.C. initiated the project and wrote the manuscript draft. W.G. and M.W. carried out the uptake assay, purified the protein, prepared the cryo-EM samples, and collected the cryo-EM data, processed the data and built the model. J.X. helped to collect the cryo-EM data. W.G., J.Z., Y.L., and Y.C. generated anti-hURAT1 antibodies. All authors contributed to the manuscript preparation.

## Competing interests

The authors declare no competing interests.
