## [Peer Review file · Nature Communications]

Mechanisms of urate transport and uricosuric drugs inhibition in human URAT1

Corresponding Author: Dr Lei Chen

Version 0:

Reviewer comments:

Reviewer #1

(Remarks to the Author)

URAT1 is a key transporter for renal urate reabsorption, playing a vital role in maintaining urate balance within the body. It is one of the drug targets for treating hyperuricemia (high urate levels). In this study, the authors present three structures of URAT1 in complex with different molecules: urate (the substrate), benzbromarone, and verinurad (both inhibitors). These structures reveal two distinct open states of the transporter: an outward-facing partially occluded state when bound to urate and an inward-facing state when bound to either inhibitor. Through mutagenesis experiments and urate uptake assays, the authors further identified key residues involved in substrate/inhibitor binding and in stabilizing the different conformations of URAT1 during the transport cycle. Although similar structural studies of URAT1 have been recently published, this work provides valuable insights into the specific mechanisms of substrate and inhibitor interactions with URAT1.

Specific comments:

1. Understanding the structure of URAT1 and how uricosuric drugs inhibit its function is of great interest. Given the recent surge in publications on URAT1 structures, including papers in *Cell Research* and *Cell Reports* by Dai and Lee and He et al., respectively, and two new preprints on bioRxiv, a detailed comparison of the structures presented in this manuscript with those already available in the PDB is essential. This comparison should highlight similarities and differences, particularly for structures bound with the same drug or substrate. Notably, the urate binding pose observed in this study appears to differ from previously reported structures, warranting further discussion and investigation to understand the potential implications of this variation.

2. The inherent difficulties in purifying wild-type URAT1 for structural studies, due to its instability during overexpression and purification, have led to various protein modifications to facilitate the purification process. Recent reports, including this study, have employed strategies such as the R477S mutation, replacing the intracellular loop with that of OAT4, and consensus mutagenesis. However, these modifications appear to have varying impacts on URAT1 transport activity and drug sensitivity. It is crucial to understand the minimal modifications necessary to achieve a stable protein for structural analysis while preserving its native functional characteristics. A critical analysis of the trade-offs between stability and functional integrity will be beneficial for future research in this area. Can the authors discuss what minimum mutations are required for stabilizing URAT1 without disturbing its function?

3. The SLC22 transporter family is generally characterized by broad substrate specificity, enabling the transport of a wide range of molecules. However, URAT1 stands out as a distinct member exhibiting remarkable specificity for urate. Do the structural features of URAT1 revealed in this study, in comparison to available structures of other SLC22 members, provide insights into the molecular basis for this difference in substrate specificity? A comparative analysis that considers key binding site residues may shed light on URAT1's unique substrate recognition properties.

Reviewer #2

(Remarks to the Author)

In this manuscript, Guo et al determined the cryo-EM structures of the human urate transporter 1 (URAT1) bound to urate and two inhibitors--benzbromarone and verinurad. The manuscript is well-written and clear, and the results appear to support the authors' discussion points. With that said, this reviewer is concerned about the potential impact of the 15 mutations

introduced in the hURAT1EM on data analysis and interpretation. Further, in light of recent similar publications, the authors should discuss the overall impact and novelty of their work in contributing to the new understanding of the transport mechanisms of URAT1.

Specific Comments:

1. To enhance protein expression, 15 mutations were introduced to WT URAT1 for cryo-EM structure determination. Some of the mutations seemed to locate close to the key residues interacting with the substrate and/or inhibitors. The potential impact of these mutations on the structural interpretation of data must be addressed. For instance, M214 was predicted to have hydrophobic interaction with verinurad based on the cryo-EM of hURAT1EM but the M214S mutant showed no change in IC50 compared to WT hURAT1. Is it possible that the prediction was based on hURAT1EM but not WT hURAT1?
2. The IC50 value of hURAT1EM for verinurad is about 3 times that of WT hURAT1, suggesting that the binding affinity might be different between the two transporters. Further, the author should determine and compare the Km values of urate for WT and hURAT1EM. This is important since the authors used structural data determined from hURAT1EM to infer functional information for the WT hURAT1.
3. The authors mentioned that the F365A mutation likely increased the Km value for hURAT1. This should be verified experimentally in their systems.
4. For inhibition experiments, did the authors select a urate concentration below the Km of urate for URAT1 as recommended for IC50 determination?
5. The authors mentioned that loss-of-function mutations in hURAT1 lead to idiopathic renal hypouricemia. If these specific mutations are known, how does this relate to the structure-function relationship of hURAT1?
6. As acknowledged by the authors in the Introduction, URAT1 functions as an anion exchanger. Can any insights be gained regarding the counter ion binding from the structures determined in this study?
7. Two recent papers have described the structures of human URAT1. How does this manuscript contribute to and build upon prior work?
 - He J, Liu G, Kong F, Tan Q, Wang Z, Yang M, He Y, Jia X, Yan C, Wang C, Qian H. Structural basis for the transport and substrate selection of human urate transporter 1. Cell Rep. 2024 Aug 27;43(8):114628. doi: 10.1016/j.celrep.2024.114628. Epub 2024 Aug 14. PMID: 39146184.
 - Dai Y, Lee CH. Transport mechanism and structural pharmacology of human urate transporter URAT1. Cell Res. 2024 Sep 9. doi: 10.1038/s41422-024-01023-1. Epub ahead of print. PMID: 39245778

Version 1:

Reviewer comments:

Reviewer #1

(Remarks to the Author)

In the revised manuscript, the authors have addressed the comments from the reviewer.

Reviewer #2

(Remarks to the Author)

The authors have adequately addressed my previous concerns and made efforts to compare their results with two recent published models in the revised manuscript.

We greatly appreciate the time, efforts, and constructive comments of reviewers for improving our manuscript. We have provided our pdb files and maps in the revision. Please see the point-to-point responses below (colored in dark blue).

Reviewer #1 (Remarks to the Author):

URAT1 is a key transporter for renal urate reabsorption, playing a vital role in maintaining urate balance within the body. It is one of the drug targets for treating hyperuricemia (high urate levels). In this study, the authors present three structures of URAT1 in complex with different molecules: urate (the substrate), benzbromarone, and verinurad (both inhibitors). These structures reveal two distinct open states of the transporter: an outward-facing partially occluded state when bound to urate and an inward-facing state when bound to either inhibitor. Through mutagenesis experiments and urate uptake assays, the authors further identified key residues involved in substrate/inhibitor binding and in stabilizing the different conformations of URAT1 during the transport cycle. Although similar structural studies of URAT1 have been recently published, this work provides valuable insights into the specific mechanisms of substrate and inhibitor interactions with URAT1.

Specific comments:

1. Understanding the structure of URAT1 and how uricosuric drugs inhibit its function is of great interest. Given the recent surge in publications on URAT1 structures, including papers in *Cell Research* and *Cell Reports* by Dai and Lee and He et al., respectively, and two new preprints on bioRxiv, a detailed comparison of the structures presented in this manuscript with those already available in the PDB is essential. This comparison should highlight similarities and differences, particularly for structures bound with the same drug or substrate. Notably, the urate binding pose observed in this study appears to differ from previously reported structures, warranting further discussion and investigation to understand the potential implications of this variation.

Response: We are hesitant to compare our structures to the one published on BioRxiv (Wu et al., 2024) due to concerns regarding the quality of their map. Additionally, the PDB files and maps from the other study on BioRxiv (Suo et al., 2024) are currently unavailable. We have provided detailed comparisons with the structures described in the published works (Dai and Lee, 2024; He et al., 2024).

Notably, the urate-binding poses differ significantly between our findings and those reported by others (Dai and Lee, 2024) (Supplementary Fig. 10f, g). To support our structural model, we have included the electron density of urate contoured at multiple levels in Supplementary Fig. 4b. As shown in the maps, the three protrusions on the electron density match the features of urate very well. The reason for this discrepancy between two studies remains unclear; however, we speculate that the difference in pH during cryo-EM sample preparation may be a factor. They utilized a pH of 8.5, while we employed a pH of 7.4, which is closer to physiological conditions. We have discussed this difference in lines 233-239. Furthermore, the 8' oxygen groups of urate

are missing in all of electron densities from Dai and Lee (PDB ID: 9B1I, 9B1K, 9B1J) (Dai and Lee, 2024) and the densities match close related xanthine molecule (Figure R1). But the exact reason for this is unknown.

Figure R1. a, The chemical structure of urate. b, Electron density map (blue mesh, contoured at 5σ) of urate in this study. c, The chemical structure of xanthine. d-f, Electron density maps (blue mesh, contour levels shown in the parenthesis) of urate from Dai and Lee.

2. The inherent difficulties in purifying wild-type URAT1 for structural studies, due to its instability during overexpression and purification, have led to various protein modifications to facilitate the purification process. Recent reports, including this study, have employed strategies such as the R477S mutation, replacing the intracellular loop with that of OAT4, and consensus mutagenesis. However, these modifications appear to have varying impacts on URAT1 transport activity and drug sensitivity. It is crucial to understand the minimal modifications necessary to achieve a stable protein for structural analysis while preserving its native functional characteristics. A critical analysis of the trade-offs between stability and functional integrity will be beneficial for future research in this area. Can the authors discuss what minimum mutations are required for stabilizing URAT1 without disturbing its function?

Response: The study by He et al. (He et al., 2024) utilized the loss-of-function mutation R477S, resulting in a URAT1 protein that is essentially nonfunctional. In contrast, Dai and Lee (Dai and Lee, 2024) replaced the extracellular loop residues 60-65 and intracellular residues 280-343 of URAT1 with those of OAT4, mutating 70 residues while maintaining URAT1's functionality. Another study on BioRxiv generated 53 mutations, resulting in a URAT1 protein with little function (Suo et al., 2024). Additionally, one study on BioRxiv did not introduce any mutations, but the map quality was low (Wu et al., 2024). In our consensus design, we excluded residues near the central substrate binding site and introduced 15 mutations to create a functional URAT1 protein. We summarized these studies in Table R1.

Table R1. The summary of recent papers related to the structures of hURAT1.

Citation	Mutations	Transport Activity (Compared to wild-type URAT1)	Resolution and map quality
Jingjing He, et al. Cell Reports 43, 114628, August 27, 2024.	1 mutation (R477S)	Reduced	3.0 Å
Yaxin Dai, Chia-Hsueh Lee. Cell Research (2024) 34:776–787.	70 mutations (Residues: 60-65 aa and 280–343 aa)	Similar	2.7 -3.73 Å
This study	15 mutations (A162V, M186S, A195T, A226S, V264A, W294R, W300R, G309R, M330V, P333A, M343T, F348L, M383V, F384L, H402R)	Robust transporting activity	3.0 -3.3 Å
Canrong Wu, et al. bioRxiv. September 11, 2024.	Wild-type URAT1		3.2 -4.7 Å (low map quality)
Yang Suo, et al. bioRxiv. September 12, 2024.	53 mutations (L9Q, M22V, M25V, S27P, A51V, I63V, L64P, S66A, S68G, I75V, R84G, S107N, D112A, I126T, N136D, H142Q, A162V, M186S, A195T, L206V, L221V, A226S, R228Q, G246V, T248M, T259A, L274V, W277C, T288I, W294R, Q297R, W300Q, G309R, Q312G, E318Q, R325Q, M330V, P333A, M343T, F348L, M383V, F384L, M394I, A396T, H402R, L409Q, L414V, V445L, S457G, L523A, L543G, G544N, N545P)	Reduced	2.6 -3.0 Å

During our studies, we attempted to reintroduce consensus design mutations back into the wild-type URAT1 construct and found that the expression level of URAT1 increased with the number of mutations introduced; however, none of the individual mutations significantly improved the behavior of URAT1. This suggests that the enhancement of URAT1's expression and functionality is a composite effect of these mutations (Figure R2). Consequently, we proceeded with the optimal URAT1_{EM} for our studies.

Figure R2. Expression levels of URAT1 mutants.

3. The SLC22 transporter family is generally characterized by broad substrate specificity, enabling the transport of a wide range of molecules. However, URAT1 stands out as a distinct member exhibiting remarkable specificity for urate. Do the structural features of URAT1 revealed in this study, in comparison to available structures of other SLC22 members, provide insights into the molecular basis for this difference in substrate specificity? A comparative analysis that considers key binding site residues may shed light on URAT1's unique substrate recognition properties.

Response: While it is well-known that URAT1 can transport urate, it is also capable of transporting a variety of other substrates, including nicotinate and pyrazinecarboxylic acid (Enomoto et al., 2002). Given the limited available structures, we feel it is too early to draw definitive conclusions about the substrate specificity of URAT1. However, we can leverage information from existing structures and sequence alignments to understand why URAT1 does not transport certain substrates associated with other SLC22 transporters.

To investigate this, we compared the structures of hURAT1 with those of other SLC22 family members. Our analysis revealed significant differences in two residues within the substrate binding site. Specifically, bulky F360 is substituted by smaller residues such as T, V, or L in OATs, and by D or S in OCTs. Additionally, long charged K393 is replaced by a shorter alanine residue in OCTs (Supplementary Fig. 2 and Figure R3). Further structural superposition provided insights into the implications of these differences. For example, the OCT1 substrates diltiazem and metformin clash with residues F360 and F365; the hOCT1 substrate fenoterol clashes with

F360 and F449; thiamine clashes with F360, F365, and K393 of URAT1; the hOCT2 substrate 1-methyl-4-phenylpyridinium (MPP⁺) clashes with K393 and R477; and the rOAT1 substrate α -ketoglutaric acid (α -KG) clashes with S238 and K393 (Figure R4). This analysis elucidates why hURAT1 does not transport diltiazem, metformin, fenoterol, thiamine, MPP⁺, and α -KG.

Figure R3. Sequence alignment among hURAT1, rOAT1 and hOCTs. The sequence alignment of human URAT1 (hURAT1), rat OAT1 (rOAT1), human OCT1 (hOCT1), human OCT2 (hOCT2), and human OCT3 (hOCT3). Conserved residues are shaded in grey. Secondary structures are shown above and colored in blue. Residues clashed with diltiazem, fenoterol, metformin, thiamine, MPP⁺ and α -KG are denoted by * and colored in green, red, green, purple, orange and blue, respectively.

Figure R4. Structure comparisons among hURAT1s, substrates-bound hOCT1/2 and substrate-bound rOAT1. a-d, The structures of substrates-bound hOCT1 are overlaid onto the benzbromarone-bound hURAT1 (yellow). Diltiazem-bound (a), fenoterol-bound (b), metformin-bound (c) and thiamine-bound (d) hOCT1s are colored in green, light blue, pink and slate, respectively. e, The structure of MPP⁺-bound hOCT2 (orange) is overlaid onto the urate-bound hURAT1 (cyan). MPP⁺, 1-methyl-4-phenylpyridinium. f, The structure of α-KG-bound rOAT1 (palegreen) is overlaid onto the benzbromarone-bound hURAT1 (yellow). α-KG, α-ketoglutaric acid. Substrates and related residues in hURAT1 are shown as sticks. Steric clashes between substrates and residues are denoted with red dashed circles.

Reviewer #2 (Remarks to the Author):

In this manuscript, Guo et al determined the cryo-EM structures of the human urate transporter 1 (URAT1) bound to urate and two inhibitors--benzbromarone and verinurad. The manuscript is well-written and clear, and the results appear to support the authors' discussion points. With that said, this reviewer is concerned about the potential impact of the 15 mutations introduced in the hURAT1EM on data analysis and interpretation. Further, in light of recent similar publications,

the authors should discuss the overall impact and novelty of their work in contributing to the new understanding of the transport mechanisms of URAT1.

Response: While our work was under submission to several journals, two other studies (Dai and Lee, 2024; He et al., 2024) were published. He et al. determined the structure of the R477S loss-of-function mutant, while Dai et al. elucidated several structures of URAT1 in complex with substrates and inhibitors. In comparing our work with Dai's, our structure with urate bound is at a physiological pH of 7.5, whereas they conducted their experiments at pH 8.5, resulting in an inverted binding pose for urate. Additionally, we provided the structure of URAT1 in complex with benzbromarone, which was absent in the other studies.

Specific Comments:

1. To enhance protein expression, 15 mutations were introduced to WT URAT1 for cryo-EM structure determination. Some of the mutations seemed to locate close to the key residues interacting with the substrate and/or inhibitors. The potential impact of these mutations on the structural interpretation of data must be addressed. For instance, M214 was predicted to have hydrophobic interaction with verinurad based on the cryo-EM of hURAT1EM but the M214S mutant showed no change in IC₅₀ compared to WT hURAT1. Is it possible that the prediction was based on hURAT1EM but not WT hURAT1?

Response: None of the 15 mutations in our URAT1_{EM} are located near the substrate or inhibitor binding site. The closest mutation is M186S, which is 13 Å away from verinurad. Moreover, the binding pose of verinurad in our structure is highly similar to the one determined by others (PDB ID: 9B1I) (Supplementary Fig. 10), providing additional support for the accuracy of our model.

2. The IC₅₀ value of hURAT1EM for verinurad is about 3 times that of WT hURAT1, suggesting that the binding affinity might be different between the two transporters. Further, the author should determine and compare the K_m values of urate for WT and hURAT1EM. This is important since the authors used structural data determined from hURAT1EM to infer functional information for the WT hURAT1.

Response: We have provided the K_m value in the revised Fig. 1b.

3. The authors mentioned that the F365A mutation likely increased the K_m value for hURAT1. This should be verified experimentally in their systems.

Response: K_m of F365A was measured previously. Please see Table 1 in (Zhao et al., 2020).

4. For inhibition experiments, did the authors select a urate concentration below the K_m of urate for URAT1 as recommended for IC₅₀ determination?

Response: Due to the low signal to noise ratio of urate uptake assay at low urate concentration, we used 400 μM for inhibition assay throughout the study, which is slightly higher than the K_m

of urate as shown in the revised Fig. 1b.

5. The authors mentioned that loss-of-function mutations in hURAT1 lead to idiopathic renal hypouricemia. If these specific mutations are known, how does this relate to the structure-function relationship of hURAT1?

Response: Several URAT1 mutations have been identified in patients with idiopathic renal hypouricemia. Among these, the R477 mutation directly impacts urate binding. We have included a discussion on this in the revised manuscript, specifically in lines 107-109. Additionally, the M430T mutation leads to reduced surface expression, while the W258X and Q297X mutations result in non-functional truncated proteins (Ichida et al., 2004). We chose not to include these mutations in our manuscript.

6. As acknowledged by the authors in the Introduction, URAT1 functions as an anion exchanger. Can any insights be gained regarding the counter ion binding from the structures determined in this study?

Response: In the literature (Enomoto et al., 2002), several counter ions, including lactate, have been proposed. However, we were unable to reproduce the results indicating that lactate inhibits urate uptake (Figure R5). The concentrations of other metabolites, such as nicotinate and probenecid (Enomoto et al., 2002), inside the epithelial cells of the proximal tubule may not be sufficient to support a continuous exchange of urate from glomerular filtrate. Consequently, the identity of the endogenous counter ions of URAT1 remains elusive, and we have not yet pursued the study of the mechanism of counter ion binding.

Figure R5. Urate uptake activity of hURAT1_{WT} in the presence or absence of lactate.

7. Two recent papers have described the structures of human URAT1. How does this manuscript contribute to and build upon prior work?

- He J, Liu G, Kong F, Tan Q, Wang Z, Yang M, He Y, Jia X, Yan C, Wang C, Qian H. Structural basis for the transport and substrate selection of human urate transporter 1. Cell Rep. 2024 Aug 27;43(8):114628. doi: 10.1016/j.celrep.2024.114628. Epub 2024 Aug 14. PMID: 39146184.

- Dai Y, Lee CH. Transport mechanism and structural pharmacology of human urate transporter URAT1. *Cell Res.* 2024 Sep 9. doi: 10.1038/s41422-024-01023-1. Epub ahead of print. PMID: 39245778

Response: While our work was under submission to several journals, two other studies (Dai and Lee, 2024; He et al., 2024) were published. He et al. determined the structure of the R477S loss-of-function mutant, while Dai et al. elucidated several structures of URAT1 in complex with substrates and inhibitors. In comparing our work with Dai's, our structure with urate bound is at a physiological pH of 7.5, whereas they conducted their experiments at pH 8.5, resulting in an inverted binding pose for urate. Additionally, we provided the structure of URAT1 in complex with benzbromarone, which was absent in the other studies.

References:

- Dai, Y., and Lee, C.H. (2024). Transport mechanism and structural pharmacology of human urate transporter URAT1. *Cell Res.* 34, 776-787.
- Enomoto, A., Kimura, H., Chairoungdua, A., Shigeta, Y., Jutabha, P., Cha, S.H., Hosoyamada, M., Takeda, M., Sekine, T., Igarashi, T., *et al.* (2002). Molecular identification of a renal urate anion exchanger that regulates blood urate levels. *Nature* 417, 447-452.
- He, J., Liu, G., Kong, F., Tan, Q., Wang, Z., Yang, M., He, Y., Jia, X., Yan, C., Wang, C., and Qian, H. (2024). Structural basis for the transport and substrate selection of human urate transporter 1. *Cell Rep* 43, 114628.
- Ichida, K., Hosoyamada, M., Hisatome, I., Enomoto, A., Hikita, M., Endou, H., and Hosoya, T. (2004). Clinical and molecular analysis of patients with renal hypouricemia in Japan-influence of URAT1 gene on urinary urate excretion. *J. Am. Soc. Nephrol.* 15, 164-173.
- Suo, Y., Fedor, J.G., Zhang, H., Tsoolova, K., Shi, X., Sharma, K., Kumari, S., Borgnia, M., Zhan, P., Im, W., and Lee, S.-Y. (2024). Molecular basis of the urate transporter URAT1 inhibition by gout drugs. *bioRxiv*, 2024.2009.2011.612563.
- Wu, C., Zhang, C., Jin, S., Wang, J.J., Dai, A., Xu, J., Zhang, H., Yang, X., He, X., Yuan, Q., *et al.* (2024). Molecular mechanisms of uric acid transport by the native human URAT1 and its inhibition by anti-gout drugs. *bioRxiv*, 2024.2009.2011.612394.
- Zhao, Z., Jiang, Y., Li, L., Chen, Y., Li, Y., Lan, Q., Wu, T., Lin, C., Cao, Y., Nandakumar, K.S., *et al.* (2020). Structural Insights into the Atomistic Mechanisms of Uric Acid Recognition and Translocation of Human Urate Anion Transporter 1. *ACS Omega* 5, 33421-33432.

We greatly appreciate the time, efforts, and constructive comments of reviewers for improving our manuscript.

REVIEWERS' COMMENTS

Reviewer #1 (Remarks to the Author):

In the revised manuscript, the authors have addressed the comments from the reviewer.

Reviewer #2 (Remarks to the Author):

The authors have adequately addressed my previous concerns and made efforts to compare their results with two recent published models in the revised manuscript.